# Industrial Ceramic Brick Drying in Oven by CFD

**DOI:** 10.3390/ma12101612

**Published:** 2019-05-16

**Authors:** Morgana de Vasconcellos Araújo, Antonildo Santos Pereira, Jéssica Lacerda de Oliveira, Vanderson Alves Agra Brandão, Francisco de Assis Brasileiro Filho, Rodrigo Moura da Silva, Antonio Gilson Barbosa de Lima

**Affiliations:** 1Postgraduate Program in Process Engineering, Federal University of Campina Grande, Campina Grande 58429-900, Paraíba, Brazil; jessica.lacerda07@gmail.com (J.L.d.O.); vanderson_agra@hotmail.com (V.A.A.B.); franciscobrasileirofilho@gmail.com (F.d.A.B.F.); antonio.gilson@ufcg.edu.br (A.G.B.d.L.); 2Baiano Federal Institute, Santa Inês, Bahia 47440-000, Brazil; antonildo.pereira@ifbaiano.edu.br; 3Federal Institute of Paraiba, Princesa Isabel, Paraíba 58015-020, Brazil; rodrigo.silva@ifpb.edu.br

**Keywords:** drying, brick, temperature, moisture, CFD

## Abstract

The drying process is a step of ceramic brick production which requires the control of process variables to provide a final product with a porous uniform structure, reducing superficial and volumetric defects and production costs. Computational fluid dynamics (CFD) is an important tool in this process control, predicting the drying physical phenomenon and providing data that improve the industrial efficiency production. Furthermore, research involving CFD brick drying has neglected the effects of oven parameters, limiting the analysis only to the bricks. In this sense, the aim of this work is to numerically study the hot air-drying process of an industrial hollow ceramic brick in an oven at 70 °C. The results of the water mass and temperature distributions inside the brick, as well as moisture, temperature, velocity and pressure fields of the oven drying air at different process times are shown, analyzed and compared with experimental data, presenting a good agreement.

## 1. Introduction

The process of moisture removal from a porous material is essential in the food industry as a way of increasing the aliments’ shelf life, or in the ceramic industry as a step in the brick production process.

Ceramics have been widely used by man since the Neolithic period, due to their simple method of production and their wide application in different fields. The main raw materials for the production of traditional ceramics are feldspar (mainly potassium), silicon (quartz) and clay minerals. They can present some natural or synthetic additives, to increase their processing capacity and final properties [1].

The fabrication of ceramic parts consists of hydrated clay molding, which is subsequently submitted to slow drying in a dryer or in an environment protected from solar irradiation in order to reduce the material water content. The optimal conditions of the drying process occur when the water removal is carried out, promoting the lowest moisture and temperature gradient inside the material structure and reducing defects such as cracks and deformations. The final step is the exposure of the material to high temperatures, which provides rigidity and resistance to ceramic products (firing process).

There is research that has proposed alternative ceramic brick production techniques, such as one that uses fine dust waste from the cyclones connected to a spray dryer in geopolymer brick production in ceramic tile manufacture. After physical-chemical characterization and studies of the preparation method, the authors concluded that the use of waste raw materials (except for caustic soda) resulted in a substantial reduction in the estimated brick production cost [2].

Nowadays, there are several ceramic products available in the market, such as bricks, slabs, tiles, artifacts with aesthetic value (kitchen utensils and decorative objects) and components for resisting high temperatures and compressions.

The drying of ceramic materials is a step that must be performed under controlled drying conditions, since poor drying results in low-quality products. To optimize the process, it is necessary to know the characteristics of the clay being used in the product manufacture. In this sense, there are several important clay properties that must be considered in the process, such as plasticity, consistency index, resistance, loss of water and retraction. Plasticity is the mechanical property determined by the deformation formed in a body when it is subjected to a determined force that is retained after the tension relief. Consistency index directly regulates the amount of moisture that a given sample must have in order to be properly molded. The clay resistance in its dry state is closely linked to its granulometric composition, which is influenced by the drying rate. The water removal is strongly dependent of the operating conditions (temperature, relative humidity and velocity), and the retraction phenomenon occurs because the place previously occupied by the water becomes empty during the drying process [3].

Several experimental and numerical studies have been carried out with the main objective of collecting data for the improvement of the ceramic materials drying process. Some of these studies are related to the mass diffusion coefficient, porosity, permeability, and drying kinetics [4,5,6,7,8,9]. There are still those which seek to understand the defects such as cracks caused by poor material firing [7,9,10,11,12,13].

Several mechanisms are responsible for the drying process. The intensity of these mechanisms depends on the process physical conditions and on the porous material properties. They are described as follows:
Liquid diffusion: occurs due to the concentration gradient and dominates the drying of grains and ceramic bricks [14];Vapor diffusion: occurs due to the partial vapor pressure gradient and is present during brick drying [14];Capillary diffusion: refers to the flow of liquid through capillary channels and on the solid surface due to the molecular attraction between the liquid and the solid. This mechanism can occur without heating, resulting only from pressure difference [14];Effusion flow (Knudsen): is only important under high-vacuum conditions, as in lyophilization processes, where the product is frozen under vacuum conditions and the formed ice crystals are sublimed [14].

The liquid diffusion theory is considered the simplest analysis, because it neglects other effects such as moisture evaporation and vapor transport inside the material during the drying process. This model considers, as the main mechanism, water migration based on Fick’s second law, which describes that the mass per unit area is proportional to the water concentration gradient [15]. This theory is widely used to describe the ceramic drying process presented in industry, like clay [15,16,17], ceramic tile [18] and ceramic bricks [6,7,16,19,20,21] via experimental, numerical and analytical methods.

Analytical solution methods are based on simplified considerations that allow differential equation solutions only for specific simplest cases, in which the input variables are defined for the calculation of one or more output variables. Although many engineering problems can be solved with simple equations, the results obtained by these methods may present a significant deviation from the real situation, not providing a detailed understanding of the studied phenomenon behavior [22].

On the numerical solution methods, a virtual prototype is developed, where the partial differential equations associated with the physical problem is converted into a system of equations based on a mathematical theory, such as the Finite Element Method or the Finite Volume Method. The use of numerical methods present a series of outstanding features that make them an extremely versatile and efficient for solving problems in engineering projects. Another advantage related to these techniques are their ease of parameterization, which enables a better understanding of how the different variables influence the results, the evaluation of different design conditions, and the development of an optimized configuration that meets all safety criteria and provides an improved product [22].

The use of numerical simulation has several positive points, as highlighted above. However, it shouldn’t be considered a substitute for traditional methods, which are fundamental to engineering problem solutions. Analytical and experimental results serve as a reference for the calibration of numerical parameters, ensuring a good representation of physical phenomena. In this way, numerical simulation is an important and complementary approach to analytical and experimental methods, bringing several benefits to the project and providing resources to achieve a good balance between quality, time and production cost [22].

The external fluid flow analysis in complex geometries, cannot be solved by analytical techniques alone. In these cases, Computational Fluid Dynamics (CFD) techniques have been used [3,23]. This numerical procedure has been applied in the design and optimization of drying equipment and processing techniques. In the literature, several studies can be found of drying ceramic bricks via CFD, enabling better process control and optimized production, with a reduction of losses and lower cost, avoiding the waste of raw materials [3,10,11,13,15,24]. When the mass flow rate is very slow, the phenomena of heat transfer and mass are practically diffusives, and fluid dynamics analysis is not necessary; only the Fourier and Fick’s laws are involved.

With respect to mathematical modeling and numerical simulations involving ceramic bricks, we can mention works that study the pollutant emissions originating from the burning of organic materials, as well as the absorption of pollutants by ceramic materials. A numerical study in blast stoves for ceramic brick firing and the emission of pollutants was carried out in [25]. In this process, the gas burns for 50 min, reaching a temperature of 1300 to 1350 °C. The results show a strong dependence between the nitrogen oxide emissions and the stove temperature level. Other research reported in [26] explores the possibility of carbon dioxide (CO_2_) sequestration in bricks which are rich in calcium and magnesium. In this study, three types of bricks were tested for carbonation in a reactor. The results show that the amount of carbonation is proportional to the reaction time. Acceptable carbonation and/or CO_2_ capture levels have been achieved under low pressure and temperature conditions which correspond to low energy costs and high environmental protection.

In this sense, the main objective of this work is to simulate the drying process of an industrial hollow ceramic brick in an oven by applying the CFD technique (Ansys CFX software, 15.0). The innovative aspect presented in this paper is related to the consideration of the air flow effects on the heat and mass transfer processes that occur in an oven environment.

## 2. Methodology

### 2.1. Experimental Procedure

For the mathematical modeling validation of ceramic brick drying, the moisture content and temperature data were taken from the drying experiments reported in [7] using a drying temperature (T) of 70 °C and a relative humidity (ϕ_rel_) of 7.68%. The experiment consists of an industrial brick drying in an oven, in which the drying air velocity, temperature and relative humidity are controlled. The mass and temperature of the sample were measured at 10 min intervals throughout the drying process. The brick mass was measured by using a digital balance with precision of ±1 g, model KC-01, from the Western company (Port Coquitlam, BC, Canada). The temperature of the brick was measured using an infrared thermometer with a range from −50 to 1000 °C, model TI 890, from Instrutherm (São Paulo, Brazil). The air velocity was measured using a digital vane anemometer, model AMI 300, from Instrutemp (São Paulo, Brazil), with a reading accuracy of ± 2%. The air relative humidity was measured using a thermohygrometer, model HT 208, from ICEL (Manaus, Brazil). The continuous drying experiments were complete when the brick mass reached a constant value. After this stage, in order to obtain a balanced moisture content, the sample was kept in the oven under the same temperature for 48 h. The test was performed under atmospheric pressure [7]. Figure 1 shows the clay brick used in the drying experiment.

### 2.2. Mathematical Procedure

#### 2.2.1. The Geometry and Mesh

The studied geometry domain is defined as a hollow ceramic brick positioned in the interior of an oven (see Figure 2a) at a height of 16.5 cm from the base. The kiln is 60 cm in width, 50 cm in height and 50 cm in depth. Figure 2b shows the ceramic brick with a parallelepiped shape measuring 93.36 mm on the base, 197 mm height and 200 mm length. The brick’s volume contains 8 rectangular holes measuring 3.882 and 3.643 on the sides. The external walls have a thickness of 7.1 mm and 9.4 mm, and the internal walls have a thickness of 6.3 mm and 7.88 mm, as also illustrated in this Figure.

The geometry and mesh were constructed using ICEM CFD 15.0 software. The mesh of the brick was built using the blocking mesh construction method. To reach the optimal mesh used in this work, a mesh refinement study was conducted. Four meshes were tested. The meshes contain 19,301, 66,769, 154,953 and 295,163 hexahedral elements. Numerical drying simulations were performed using only the brick domain, and after analysis of the temperature and moisture profiles, it was observed that the mesh with 66,769 elements provided results that were not modified by the increase in refinement. Based on this, a mesh of the entire oven containing the brick was constructed, i.e., a fluid domain containing a solid domain. The final hexahedral mesh constructed on the geometry had 673,625 hexahedral elements (see Figure 3a,c). Figure 3b shows the hexahedral mesh used for the porous material domain (brick).

#### 2.2.2. Governing Equations

In this section, the mass, momentum, and energy conservation equations are presented [27] for the solid and gaseous phases.
Solid phase

The water mass inside the solid domain was inserted by considering an additional variable. The transport equation for the additional variable, considering a solid domain with no motion, is given by:(1)$$\frac{\partial m}{\partial t}=\nabla \cdot \left(D_{S}\Delta m\right)$$
where *m* is the water mass inside the solid material at the instant of time t, and *D_S_* is the solid phase mass diffusion coefficient.

The heat transfer equation applied to the solid phase is given as follows:(2)$$\frac{\partial \left(\rho_{S}H_{S}\right)}{\partial t}=\nabla \cdot \left(\lambda_{S}\nabla T_{S}\right)$$
where *r*_S_ is the specific mass, *H_S_* is the enthalpy, *λ_S_* is the thermal conductivity and *T*_S_ is the solid phase temperature.

To simulate the drying process, there are some algebraic boundary and initial conditions that have to be appropriately chosen for the solid phase:Boundary conditions:
○At the interface: conservative interface flux. Details will be given about this later.Initial conditions:
○Brick temperature, *T_brick,0_*.○Water mass in the brick, *m_water,0_*.Gaseous phase

The mass conservation equation for the air in the fluid domain is given as:(3)$$\frac{\partial \rho_{f}}{\partial t}+\nabla \cdot \left(\rho_{f}\vec{U}\right)=0$$
where *r*_f_ is the air specific mass and $\vec{U}$ is the air velocity inside the oven.

The transport equation for an additional variable describing the water vapor flow in the fluid domain is:(4)$$\frac{\partial m}{\partial t}+\nabla \cdot \left(m\vec{U}\right)=\nabla \cdot \left(D_{air}\Delta m\right)$$
where *m* is the water vapor mass, $\vec{U}$ is the fluid velocity and *D_air_* is the water vapor diffusivity of water vapor considered as an ideal gas.

The momentum conservation equation is given by:(5)$$\frac{\partial \left(\rho_{f}\vec{U}\right)}{\partial t}=-\nabla \cdot \left(\rho_{f}\vec{U}\vec{U}\right)-\nabla p+\nabla \cdot \left\{\mu_{eff}\left[\nabla \vec{U}+{\left(\nabla \vec{U}\right)}^{T}\right]\right\}+{\vec{S}}_{M}$$
where ∇p is the pressure gradient, *μ* is the dynamic viscosity and ${\vec{S}}_{M}$ is the linear momentum source term.

To predict the turbulence behavior of the fluid flow in the domain, the *κ−ε* turbulence model was chosen. This model is classified as a two-equation model, and it has been considered as an industry standard model [26]. Thus, the effective fluid viscosity is given by:(6)$$\mu_{eff}=\mu +\mu_{t}$$
where *μ*_t_ is the turbulent viscosity defined as follows:(7)$$\mu_{t}=C_{\mu }\rho \frac{\kappa^{2}}{\varepsilon }$$
where *C**_μ_* is a constant value equal to 0.09, and *κ* and *ε* represent the turbulent kinetic energy and turbulent kinetic energy dissipation rate, respectively. These parameters are given as follows [28]:(8)$$\frac{\partial \left(\rho \kappa \right)}{\partial t}+\frac{\partial }{\partial x_{j}}\left(\rho U_{j}\kappa \right)=\frac{\partial }{\partial x_{j}}\left[\left(\mu +\frac{\mu_{t}}{\sigma_{\kappa }}\right)\frac{\partial \kappa }{\partial x_{j}}\right]+P_{\kappa }-\rho \varepsilon +P_{\kappa b}$$
(9)$$\frac{\partial \left(\rho \varepsilon \right)}{\partial t}+\frac{\partial }{\partial x_{j}}\left(\rho U_{j}\varepsilon \right)=\frac{\partial }{\partial x_{j}}\left[\left(\mu +\frac{\mu_{t}}{\sigma_{\varepsilon }}\right)\frac{\partial \varepsilon }{\partial x_{j}}\right]+\frac{\varepsilon }{\kappa }\left(C_{\varepsilon 1}P_{\kappa }-C_{\varepsilon 2}\rho \varepsilon +C_{\varepsilon 1}P_{\kappa b}\right)$$
where *C**_κ1_*, *C**_κ2_*, *σ**_κ_* and *σ**_ε_* are constants, and their values are: *C**_κ1_* = 1.44; *C**_κ2_* = 1.92; *σ**_κ_* = 1.0 and *σ**_ε_* = 1.3. *P**_κb_* and *P_b_* are the buoyancy forces terms.

The thermal energy model, which is suitable for low velocities, was adopted to describe the heat transfer, neglecting internal heating by viscous effects, and the source term is given as follows:(10)$$\frac{\partial \left(\rho_{f}h_{f}\right)}{\partial t}+\nabla \cdot \left(\rho_{f}\vec{U}h_{f}\right)=\nabla \cdot \left(\lambda_{f}\nabla T_{f}\right)$$
*λ_f_* is the thermal conductivity, *h_f_* is the enthalpy and *T_f_* is the air temperature.

The algebraic boundary and initial conditions of the fluid phase are:Boundary conditions:
○Airflow in the oven inlet, ${\dot{m}}_{dry\;air}$;○Vapor flow rate in the kiln inlet, ${\dot{m}}_{vapor}$;○At the interface between the solid and fluid phases was used conservative interface flux;○In the oven outlet was used an average pressure of 101,325 Pa.Initial conditions:
○Oven air temperature, *T_kiln,0_*;○Oven water vapor mass t, *m_vapor,0_*;Interface of the fluid and solid phases

Developing the mathematical modeling, the flux of variables between the domains was considered. For this, the condition of conservative interface flux at the contact regions between phases was adopted. This boundary condition means that the variable flux from one domain to another through the interphase occurs without losses, and the variable is treated as a third-order boundary condition.

In this case, diffusive flux of a substance on the solid surface is equivalent to the convective flow of the same substance in the air. Then, we can write:(11)$$-D_{air}\nabla m=h_{m}\left(m-m_{eq}\right)$$
where *D_air_* is the water mass diffusivity in the air; *m* is the instantaneous water mass at the interface; *m_eq_* is the water mass under equilibrium conditions, and *h_m_* is the convective mass transfer coefficient.

From Equation (11), the only property initialized is the mass diffusivity of water in the air.

For the temperature, the convective heat flux and the diffusive are equal at the surface:(12)$$-\lambda_{air}\nabla T=h_{C}\left(T_{eq}-T\right)$$
where λ*_air_* is the air thermal conductivity; *T* is the instantaneous temperature at the interface; *T_eq_* is the temperature under equilibrium conditions; and *h_C_* is the convective heat transfer coefficient.

#### 2.2.3. Physical-Chemical Properties

The physical-chemical properties of the porous material and the fluid domain adopted in the simulation are shown in Table 1.

#### 2.2.4. Simulation Data

Drying at a temperature of 70 °C and a relative humidity of 7.68% was chosen to be studied and the obtained results were compared with experimental data reported by [3]. The simulation is transient, with a time step of 10 s and a mean square residue of 10^−8^. The total drying time was established to be 950 min. The data used in the simulation are described in Table 2.

## 3. Results and Discussion

### 3.1. Heating and Drying Kinetics of the Brick

To show the accuracy of the adopted mathematical model, the temporal evolution of the surface temperature and average water mass of the ceramic brick were compared with experimental data. The temperature was measured at the point highlighted in Figure 2b. These comparisons are shown in Figure 4 and Figure 5.

The values of the convective heat transfer coefficients (*h_c_*) are provided based on numerical simulation at each instant of time, and an arithmetic mean value of 1.1603 W/(m^2^·K) was found for drying at 70 °C. Each time step in the numerical simulation provides the water mass gradient on the three Cartesian axis directions (*∂m/∂X*, *∂m/∂Y* and *∂m/∂Z*), and the general mean value on the surface (*∂m/∂*). By using the value of the water mass diffusivity in the air, the mean values of the convective mass transfer coefficients found for drying at 70 °C were: *h_m,X_* = 4.78 × 10^−5^ m/s; *h_m,Y_* = 4.87 × 10^−7^ m/s; *h_m,Z_* = 1.08 × 10^−7^ m/s and *h_m_* = 4.83 × 10^−4^ m/s. Furthermore, a mass diffusion coefficient D_brick_ = 9 × 10^−7^ m²/s was found using the least squares error technique.

From the analysis of these figures, we can see that a good agreement was obtained for average water mass. However, some deviations were obtained for surface temperature, which must be attributed to measuring errors and to the effect of water evaporation energy, and not considered in the mathematical modeling employed.

### 3.2. Heat and Mass Fields inside the Brick

The fields of temperature and water mass, which will be shown below, were obtained on the XZ and XY planes as shown in Figure 6. The gradients of temperature and ceramic brick water mass initially, after 10 min, after 60 min, and at the final time of drying, equivalent to 950 min, are shown in Figure 7 and Figure 8 for the *XZ* plane in *Y* = 0.1 m. Please note that the small hydric and thermal gradients appear in the first moments of the process and remain at the end. This physical situation, which is dependent on these gradient intensities, can cause defects in the ceramic product as it passes through the furnace (firing stage), such as cracks, fissures and warping.

Analyzing Figure 7, it is noted that the outer corners on the right side of the brick heat up more quickly, as they are in direct contact with the hot and dry air flux. The internal brick corners (holes) present a low heating velocity because of the low flow rate of hot air in these regions, especially in the central regions, where the brick presents a higher water content.

### 3.3. Heat and Mass Fields on the Surface of the Brick

Temperature fields were plotted on the brick surface (Figure 9) and large temperature gradients in several instants of the process were observed. Please note that the vertices on the right side of the brick were heated faster compared to the other points on the surface. Due to the direction of the air flow, the inner part of the brick remained cooler than the others throughout the drying process. A simple change of the brick position inside the kiln could change the temperature behavior on the solid surface. observed in the temperature field, there is a moisture gradient on the material surface (see Figure 10), which can be modified by changing the brick position inside the kiln.

### 3.4. Heat, Mass, Velocity and Pressure Fields of the Air inside the Oven

The hot air flowing over the brick provides its heating, since the brick is at a lower temperature level. Consequently, the air presents a temperature decrease, as shown in Figure 11 and Figure 12 at different instants of the drying process. In addition, the air flowing over the brick receives moisture from the ceramic material (see Figure 13 and Figure 14). It can be observed that after 10 min of drying, the air has a large quantity of water mass compared to at the end of the drying process, at which point the brick has a low water mass content.

The thermo-fluid dynamics for the air inside the oven directly influences the heating and drying kinetics, the heat and mass diffusion inside the ceramic brick, and the total drying time. Therefore, it is important to analyze the air velocity fields in order to evaluate the influence of the relative position between the oven air inlet and the brick, in order to achieve faster and more efficient drying conditions.

Figure 15 and Figure 16 show the air velocity field in the oven domain (fluid domain). Please note that the velocity distribution is almost the same during the drying process. This occurs due to the low air velocity values at the oven inlet. Analyzing these figures, note that the velocity decreases in the regions near the brick, characterizing the formation of boundary layers, where heat and mass exchange occurs by convection.

The velocity vector and the streamlines fields are respectively shown in Figure 17, Figure 18 and Figure 19, describing the air flow in the kiln and around the brick.

From the analysis of these figures, we can see recirculation zones in the downstream region of the brick, as expected. The presence of the brick drastically reduces the velocity in this region, where the air is almost stagnant. Due to this behavior, the pressure field in the oven is practically not affected.

The pressure field inside the oven is almost uniform (Figure 20), and this behavior is due to the low velocity of the air flowing inside the equipment. Please note that only at the upper and lower edges of the right side of the brick is it possible to see a very discreet lowering in the pressure value. When the air velocity is increased, a lower pressure zone at the left brick side can be observed, exactly in the region where the recirculation zones were presented in Figure 18.

## 4. Conclusions

Given the presented results, it is possible conclude that:

The drying and heating kinetic curves showed good agreement with the experimental data, since the liquid diffusion model was adopted, and the thermal energy used to evaporate the brick mass water was not taken into account. However, when adopting the constant mass diffusivity value, the drying curve described well the brick water mass lost throughout the drying process.

The brick temperature and water mass fields showed an asymmetrical behavior, different from the results reported by several studies investigating the drying of ceramic bricks considering only the brick domain. The results show that the assumption of constant convective heat and mass transfer coefficients on all walls of the material is an erroneous procedure. The values of the parameters are too dependent on the brick position inside the kiln.

From the air temperature field in the oven domain, it was possible to verify the heat lost by the air when flowing over the material, showing the physical coherence between the numerical simulation and the proposed mathematical model.

The water mass flow in the oven was clearly shown, illustrating the water absorption (in the vapor phase) by the drying air.

The oven air velocity field showed that the inner brick walls took a longer time to heat and dry, since the air flow through the holes of the material was almost null, as a consequence of the brick position inside the kiln.

The pressure field inside the oven is almost uniform, as a consequence of the low air velocity.

Finally, the heat and mass fields on the material surface showed an asymmetry in heating and drying, revealing that the procedure of measuring the temperature at the brick vertex plays an important role to better understanding the process.

## Figures and Tables

**Figure 1 materials-12-01612-f001:**
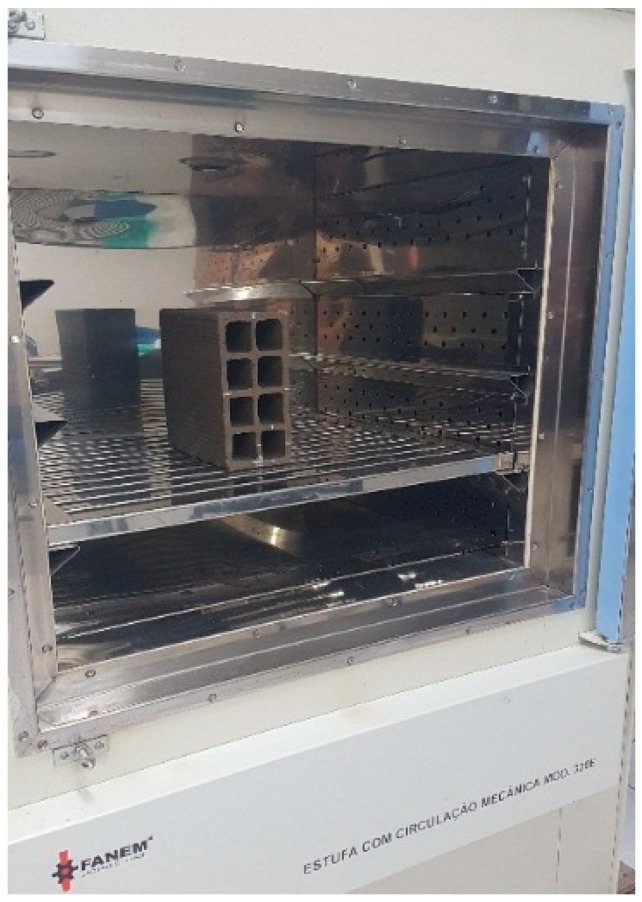
View of the ceramic brick in the drying oven (model 320E, FANEM, São Paulo, Brazil).

**Figure 2 materials-12-01612-f002:**
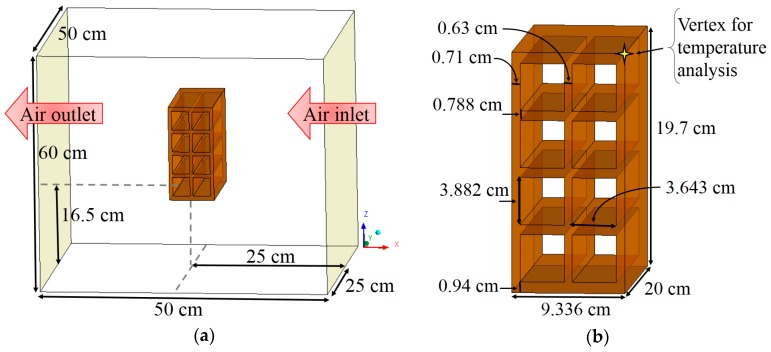
Geometrical representative of the (**a**) oven with the ceramic brick and (**b**) the brick.

**Figure 3 materials-12-01612-f003:**
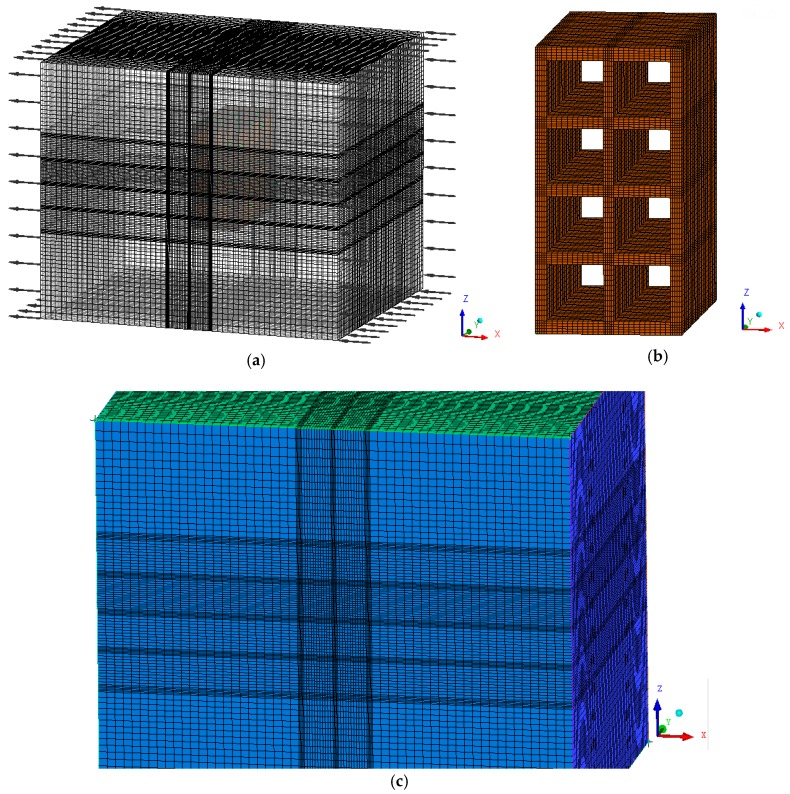
Mesh of (**a**) the full domain, (**b**) the brick, and (**c**) details of oven surface.

**Figure 4 materials-12-01612-f004:**
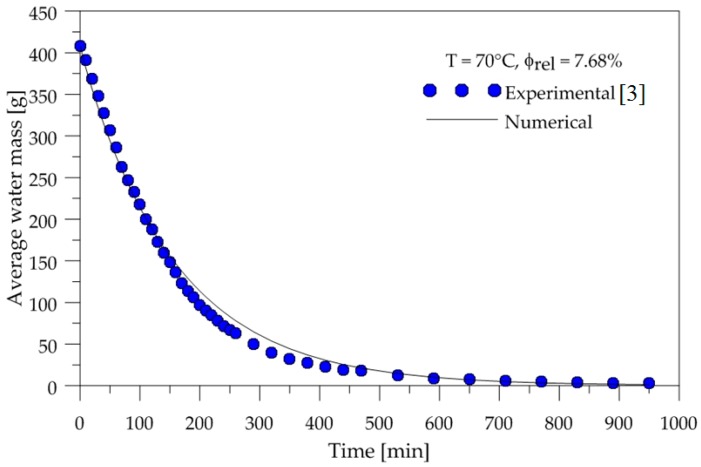
Average water mass inside the brick as a function of time.

**Figure 5 materials-12-01612-f005:**
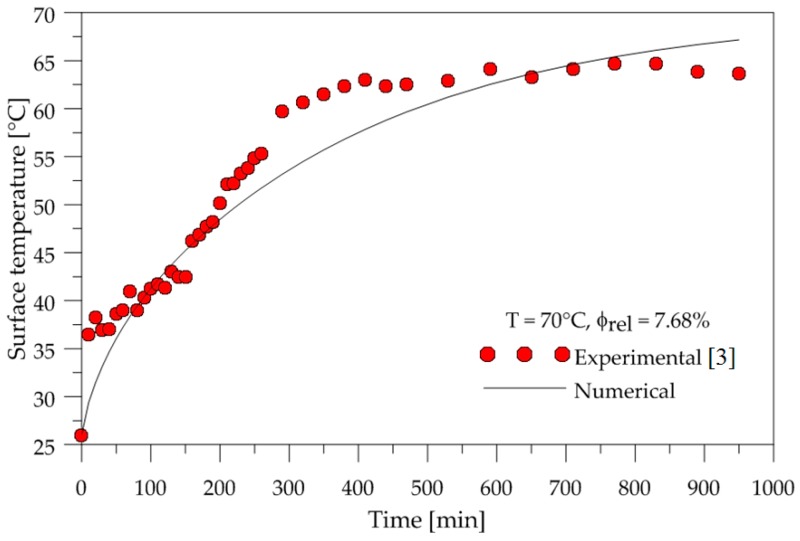
Brick surface temperature as a function of time.

**Figure 6 materials-12-01612-f006:**
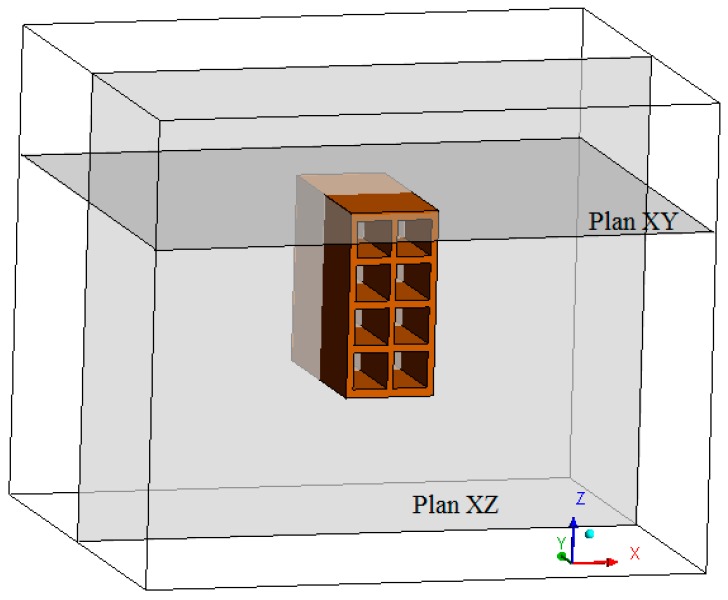
Planes XZ (Y = 0.1 m) and XY (Z = 0.196 m).

**Figure 7 materials-12-01612-f007:**
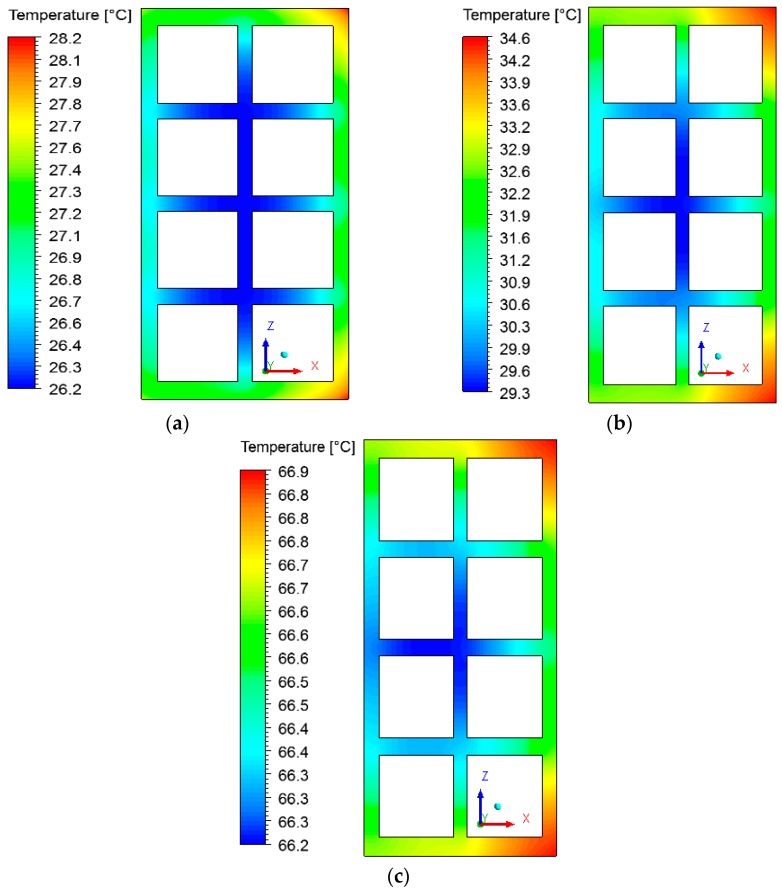
Temperature fields inside the ceramic brick at different moments of drying: (**a**) 10 min, (**b**) 60 min, and (**c**) 950 min.

**Figure 8 materials-12-01612-f008:**
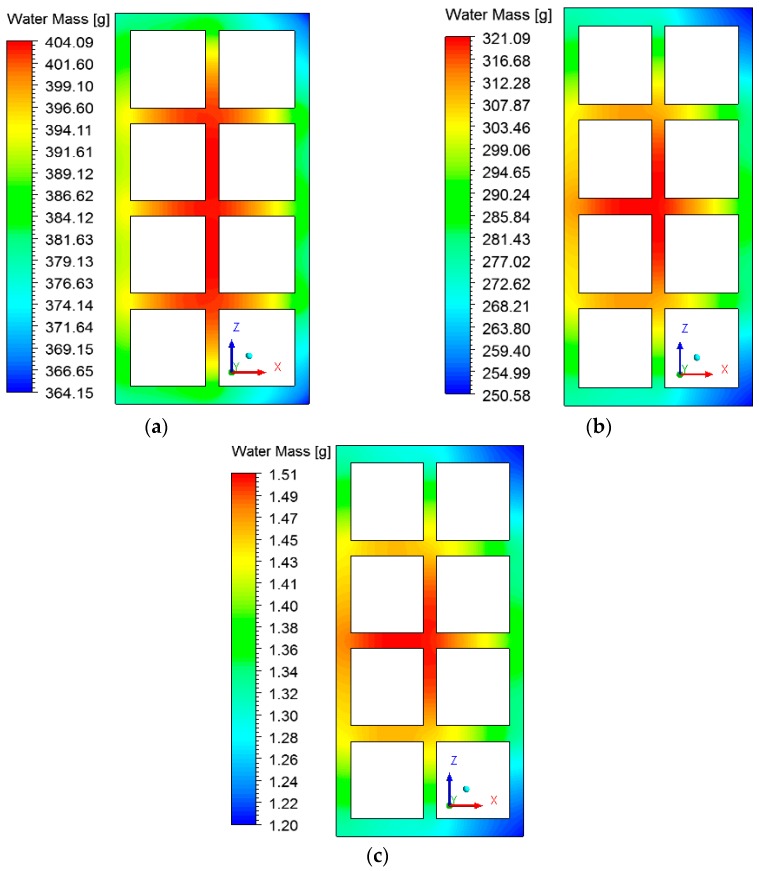
Water mass fields inside the ceramic brick at different moments of drying: (**a**) 10 min, (**b**) 60 min, and (**c**) 950 min.

**Figure 9 materials-12-01612-f009:**
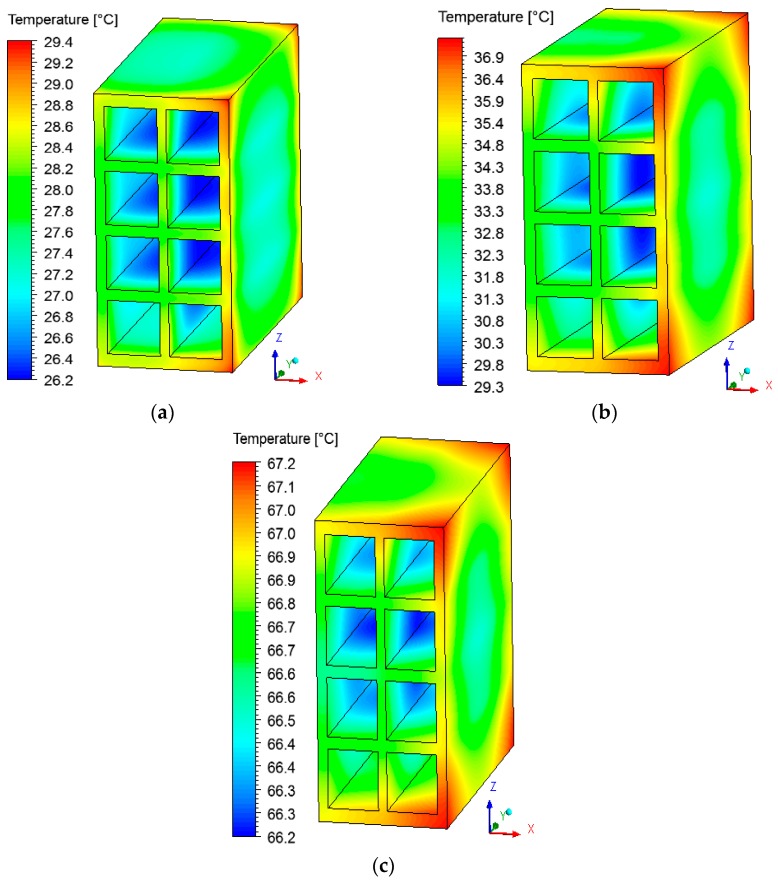
Temperature fields on the external surface of the brick at different moments of drying: (**a**) 10 min, (**b**) 60 min, and (**c**) 950 min.

**Figure 10 materials-12-01612-f010:**
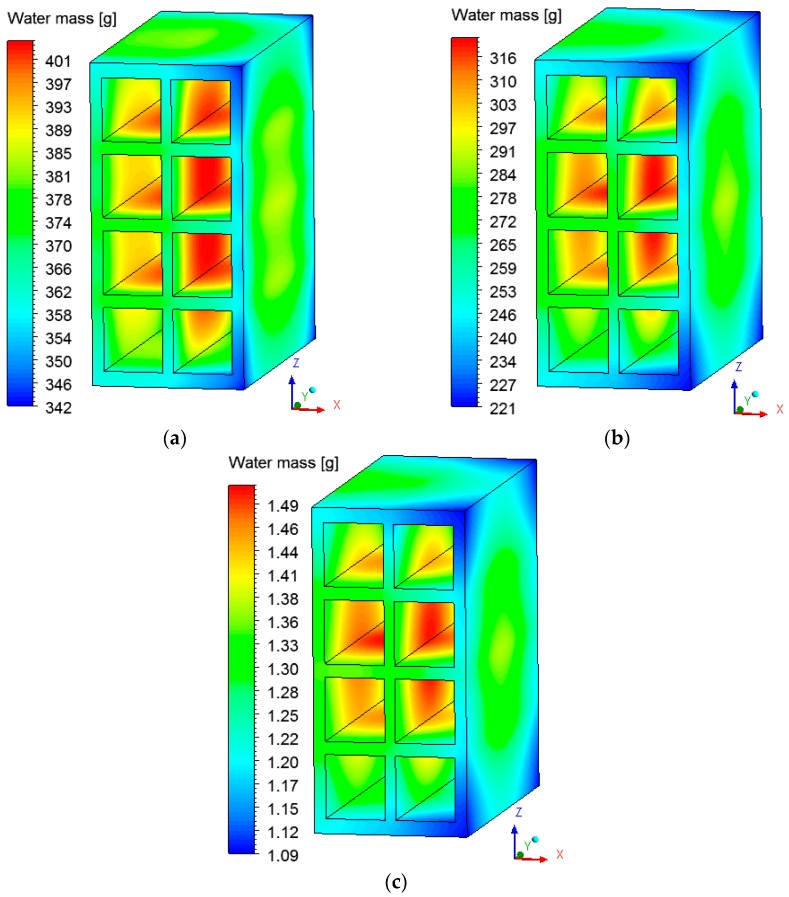
Water mass fields on the external surface of the brick at different moments of drying: (**a**) 10 min, (**b**) 60 min, and (**c**) 950 min.

**Figure 11 materials-12-01612-f011:**
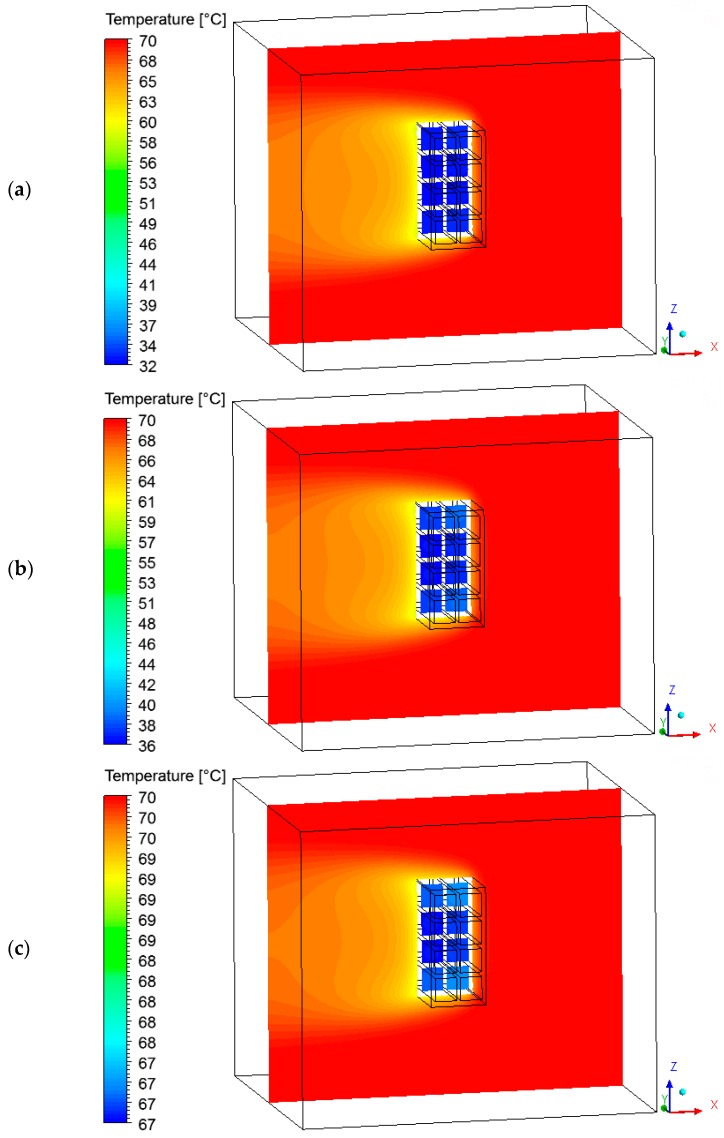
Temperature fields of the air in a plane XZ (Y = 0.2 m) at different moments of drying: (**a**) 10 min, (**b**) 60 min, and (**c**) 950 min.

**Figure 12 materials-12-01612-f012:**
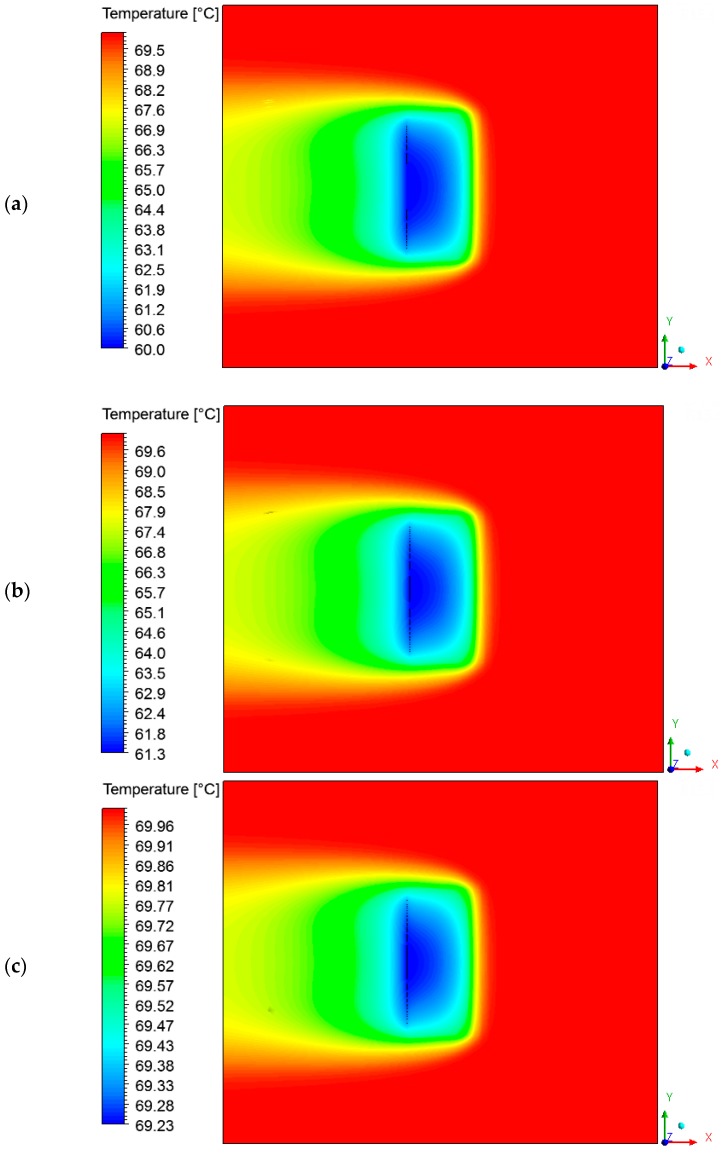
Temperature fields of the air in a plane XY (Z = 0.197 m) at different moments of drying: (**a**) 10 min, (**b**) 60 min and (**c**) 950 min.

**Figure 13 materials-12-01612-f013:**
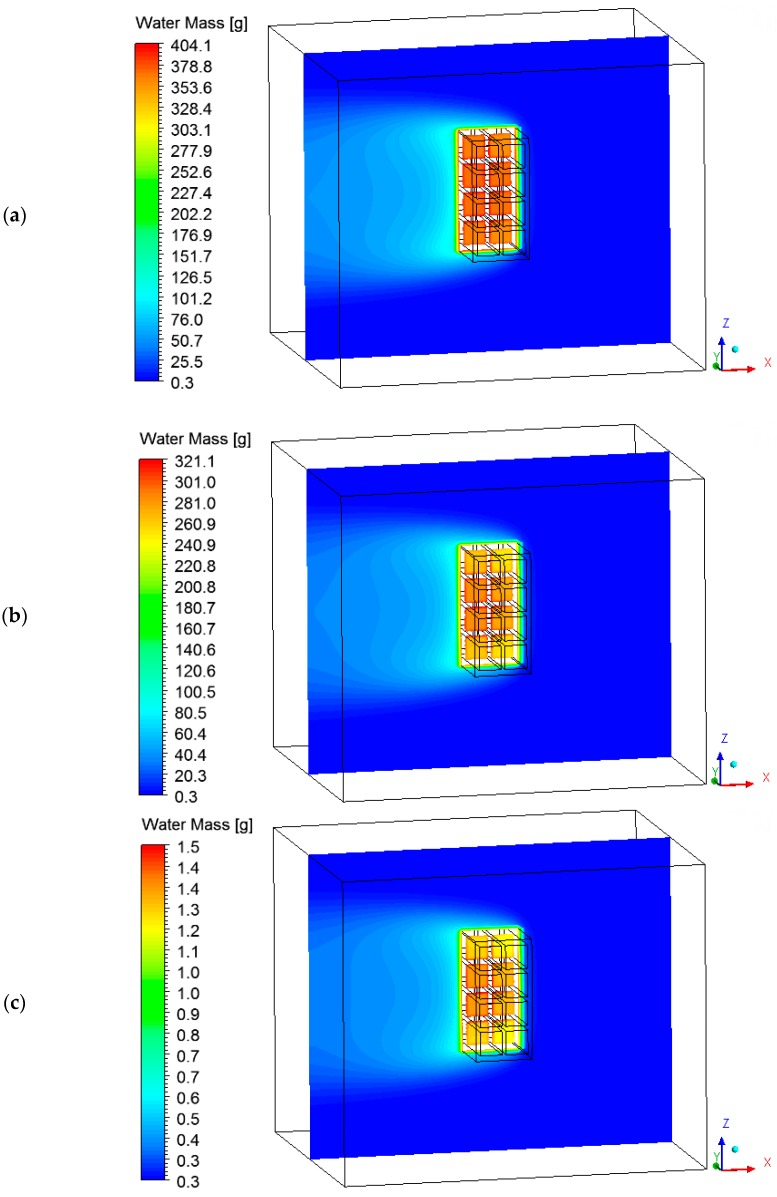
Water mass fields of the air in a plane XZ (Y = 0.1 m) at different moments of drying. (**a**) 10 min, (**b**) 60 min, and (**c**) 950 min.

**Figure 14 materials-12-01612-f014:**
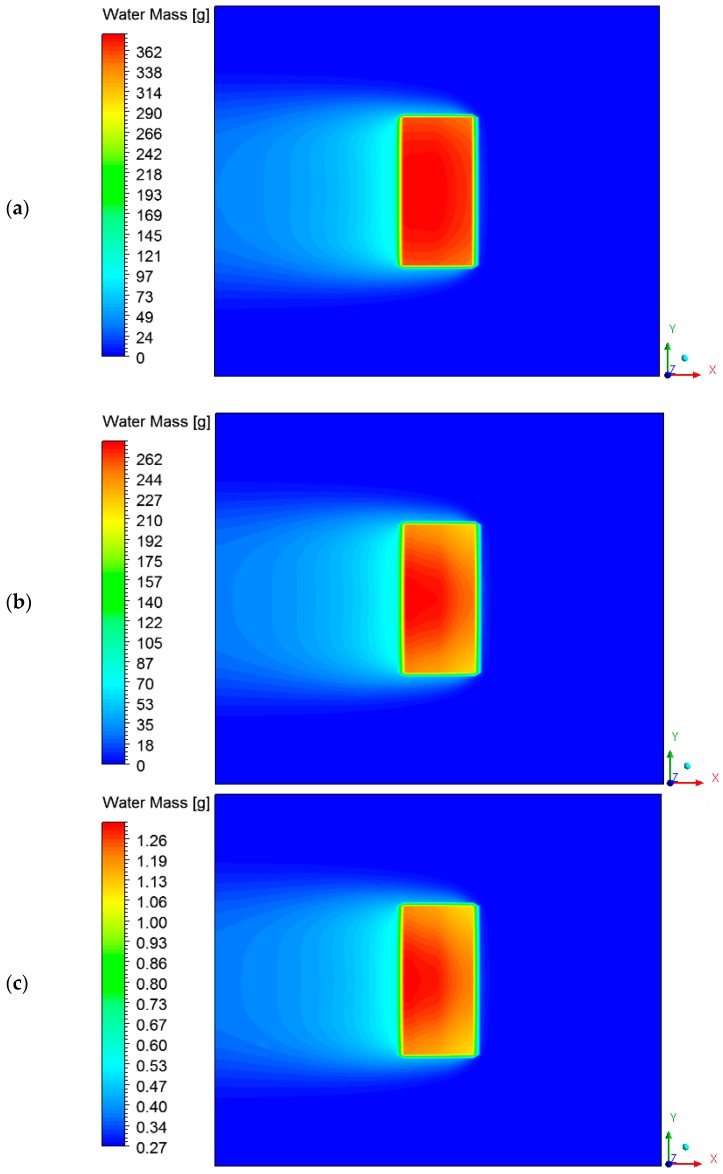
Water mass fields of the air in a plane XY (Z = 0.197 m) at different moments of drying: (**a**) 10 min, (**b**) 60 min, and (**c**) 950 min.

**Figure 15 materials-12-01612-f015:**
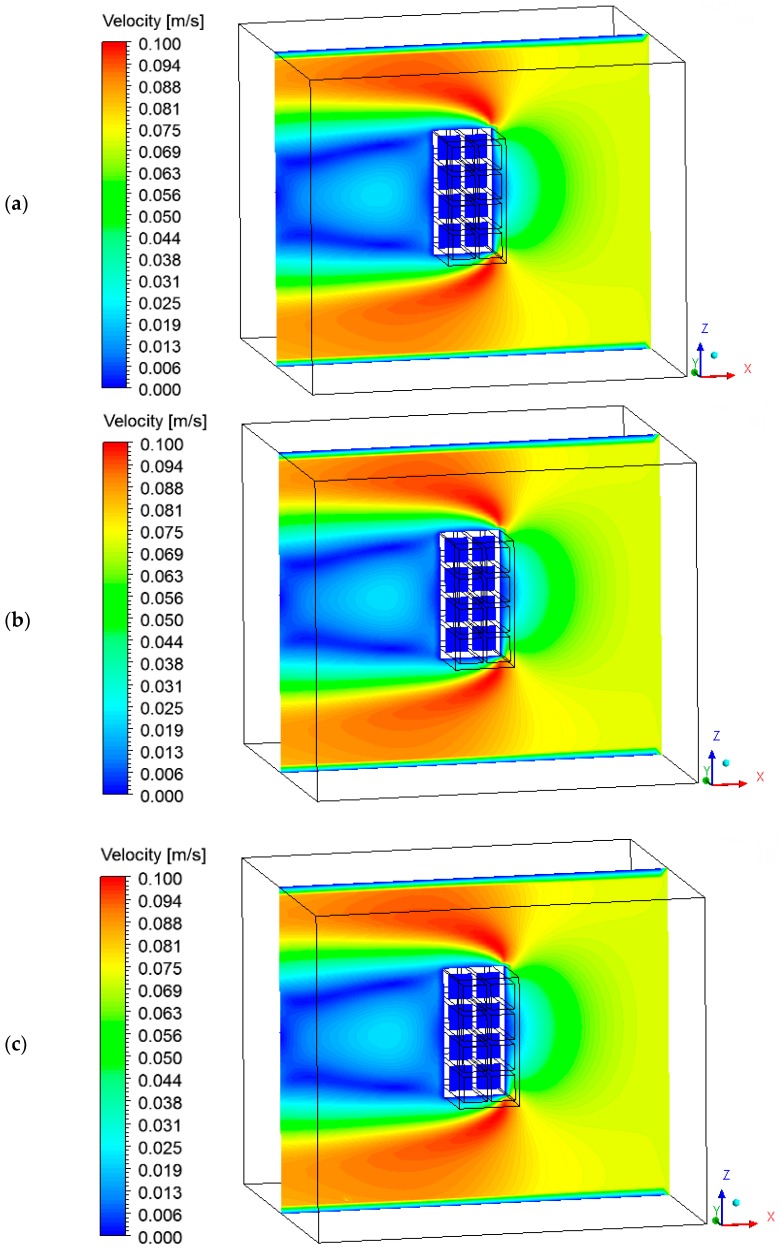
Velocity field of the air in the plane XZ (Y = 0.1 m) at different moments of drying: (**a**) 10 min, (**b**) 60 min, and (**c**) 950 min.

**Figure 16 materials-12-01612-f016:**
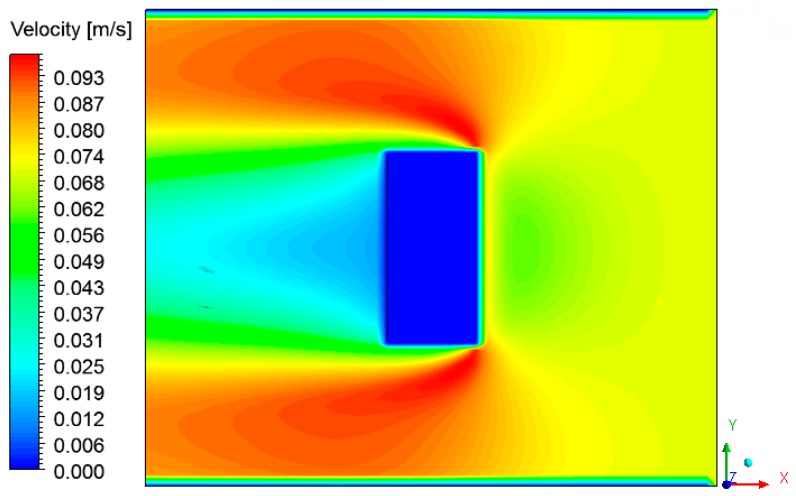
Velocity field of the air in the plane XY (Z = 0.197 m) at the instant 950 min.

**Figure 17 materials-12-01612-f017:**
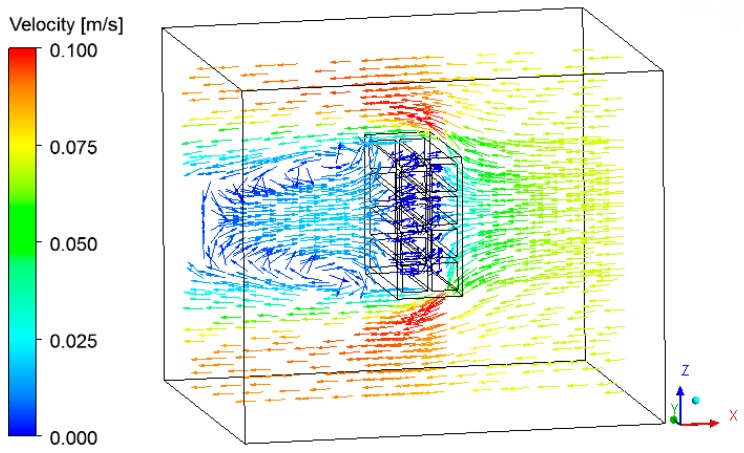
Vector field of the air in the plane XZ (Z = 0.197 m) at the instant 950 min.

**Figure 18 materials-12-01612-f018:**
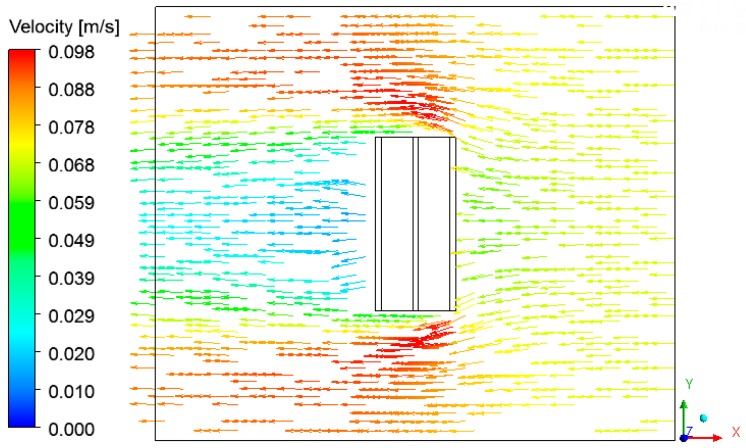
Vector field of the air in the plane XY (Z = 0.197 m) at the instant 950 min.

**Figure 19 materials-12-01612-f019:**
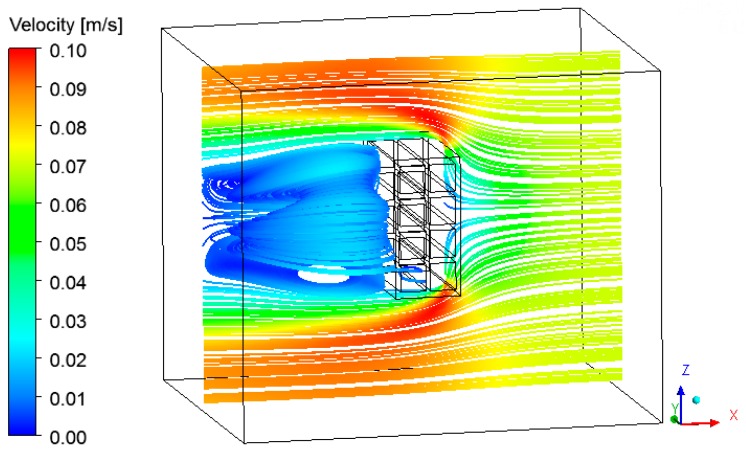
Streamlines field of the air in the plane XZ (Y = 0.1 m) at the instant 950 min.

**Figure 20 materials-12-01612-f020:**
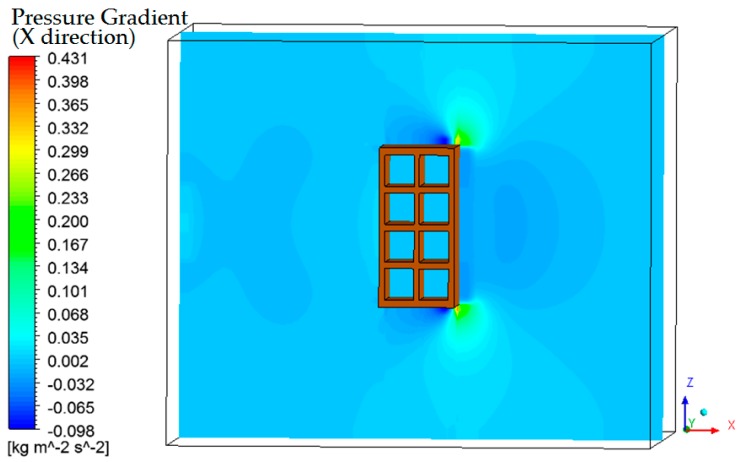
Pressure gradient field of the air in the plane XZ (Y = 0.1 m) at the instant 950 min.

**Table 1 materials-12-01612-t001:** Physical-chemical properties of the materials used in the simulations.

Property	Material	
	Solid Domain	Fluid Domain
	Clay Brick	Air
Specific mass	1985.8 kg/m³ [14]	1.029 kg/m³
Specific heat capacity	1673 J/(kg·K) [14]	1004.4 J/(kg·K)
Thermal conductivity	1.675 W/(m·K) [29]	0.0261 W/(m·K)
Dynamic viscosity	-	1.831 × 10^−5^ kg/(m·s)
Mass diffusivity	-	3.0662 × 10^−5^ m^2^/s

**Table 2 materials-12-01612-t002:** Data used in the drying simulation.

	Solid Domain		Fluid Domain
*T_brick,0_*	26 °C	*T_oven,0_*	70 °C
*m_water,0_*	408 g	*m_vapor,0_*	2.30198 g
*m_water_eq_*	1 g	${\dot{m}}_{dry\;air}$	18.00225 g/s
*m_dry_brick_*	2647 g	${\dot{m}}_{vapor}$	0.27112 g/s

where *T*_brick,0_ and *T_oven,0_* are the initial temperature of the brick and of the oven, *m_water,0_* and *m_vapor,0_* are the initial temperature of the water vapor in the brick and in the oven, *m_water_eq_* is the mass of water in in the brick in the equilibrium, *m_dry_brick_* is the mass of the dry brick, ${\dot{m}}_{dry\;air}$ is the flow of dry air entering the oven and ${\dot{m}}_{vapor}$ is the flow of water vapor entering the oven.

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
