# Peer review of "Industrial Ceramic Brick Drying in Oven by CFD"

_materials, 2019, doi:10.3390/ma12101612_

Reviewer 1 Report

The article discusses an application of CFD aimed at characterizing the drying of ceramics in an industrial oven. The problem is interesting, although the study has some limitations that need to be addressed before publication:

More than half of the abstract gives general information which should belong to the introduction. The abstract should be reformulated, focusing on the analysis and results obtained in this study.

The experimental methods are well described, but an interesting extension could be to assess the uncertainty of the measurements and the sensitivity of the various measured quantities (see the work by Rezaeiravesh et al., European Journal of Mechanics-B/Fluids 72, 57-73, 2018).

Although the approach used here based on modeling is useful, the authors should acknowledge the role of high-fidelity accurate data in applied cases. For instance, see the work by Vinuesa et al., Flow, turbulence and combustion 99 (3-4), 613-641, 2017.

How was the mesh designed? What quantity was used to assess the grid convergence? Additional justification is needed.

Is the brick at the center of the domain? Are the results comparable with the experiment, where the brick is sitting on a grid?

Equation (5): the way this equation is written, it seems that all the flow scales are solved, which is not the case. The authors should describe which RANS model (if any) they used and include the corresponding Reynolds stresses in the equation.

Figure 5: what are the sources of deviation between numerical and experimental data? The experimental result is probably more realistic, since the simulation relies heavily on modeling?

Figures 11, 12, 13… all the panels on these figures show very similar results. Can the information be condensed?

Author Response

Reply to Reviewer 1:

Dear reviewer,

Thanks for your comments and suggestions. Below there are the answer for all questions.

1.      More than half of the abstract gives general information which should belong to the introduction. The abstract should be reformulated, focusing on the analysis and results obtained in this study.

Answer: The abstract was reformulated as proposed.

2.      The experimental methods are well described, but an interesting extension could be to assess the uncertainty of the measurements and the sensitivity of the various measured quantities (see the work by Rezaeiravesh et al., European Journal of Mechanics-B/Fluids 72, 57-73, 2018). Although the approach used here based on modeling is useful, the authors should acknowledge the role of high-fidelity accurate data in applied cases. For instance, see the work by Vinuesa et al., Flow, turbulence and combustion 99 (3-4), 613-641, 2017.

Answer: In the work from which the experimental data of the ceramic brick drying were extracted, were found only the uncertainties of the measurement equipments used to obtain the process variables. Information about this topic were added to the article.

3.      How was the mesh designed? What quantity was used to assess the grid convergence? Additional justification is needed.

Answer: The mesh was designed from the blocking method, forming hexahedral elements. This element format provides better results than other formats. Additional information can be found in previous research. Further, this type of mesh allows the numerical solution to be obtained faster, since to build a good mesh in a specific domain, few elements are required as compared to a mesh of tetrahedral elements formed in the same domain. In order, to reach the mesh described in the article, several meshes were constructed with different levels of refinements and the same operating conditions. The described mesh has the minimum refinement from which there are no more significant changes in results with increasing number of elements. A small text talking about this topic was added to the article

4.      Is the brick at the center of the domain? Are the results comparable with the experiment, where the brick is sitting on a grid?

Answer: In the simulation, the brick is placed and oriented with respect to the fluid flow exactly as used in the experimental test. According to Figure 1, the metallic part in which the brick is placed was disregarded due to two factors: the first one related to the fact that during the experiments, the brick was only inserted in the oven when it was already fully heated, that is, the grid has a little influence on the heat distribution. The second relates to the difficulty in constructing a hexahedral mesh of this metallic part.

5.      Equation (5): the way this equation is written, it seems that all the flow scales are solved, which is not the case. The authors should describe which RANS model (if any) they used and include the corresponding Reynolds stresses in the equation.

Answer: In our work was used the k-e turbulence model and turbulence terms were added in the text.

6.      Figure 5: what are the sources of deviation between numerical and experimental data? The experimental result is probably more realistic, since the simulation relies heavily on modeling?

Answer: Some deviations between predicted and experimental results of the temperature at the brick surface were obtained, which must be attributed to measuring error and to the effect of water evaporation energy not considered in the modeling.

7.      Figures 11, 12, 13… all the panels on these figures show very similar results. Can the information be condensed?

Answer: This research is innovative, which results are important for the ceramic industry and that will help elucidate certain phenomena not yet fully known. Withdrawal of some of these results may compromise the quality of text and can difficult the interpretation of results. Then, it is important to show all results as reported in paper.

All modification occurred in the text are highlighted in red.

Reviewer 2 Report

The paper "Industrial ceramic brick drying in oven by CFD" written by Araújo  et al. is about the process of moisture removal from a ceramic brick. The paper suffer from the following points : 1- Figure 3a should be modified with giving the cross section of the mesh to have a better understanding of the meshing structure. 2- The data presented in "2.2.3 Physical-chemical properties" is better to present in a seperate table. 3- The brick material should be considered as a porous material not the solid. 4- "Figure 20. Pressure field" is not true as there is an inlet and outlet with pressure difference. 5- Data in "Figure 7" should be discussed. Why the corners are the hot points and inside are cold ? 6- Detail of thermal measurements of reference 2 is missing. 7- More references should be added :  https://doi.org/10.1016/j.jclepro.2014.07.028 https://doi.org/10.1016/j.conbuildmat.2017.09.052 https://doi.org/10.1016/j.conbuildmat.2018.05.285

Author Response

Dear reviewer,

Thanks for your comments and suggestions. Below there are the answer for all questions.

1.      Figure 3a should be modified with giving the cross section of the mesh to have a better understanding of the meshing structure.

Answer: Figure 3 was modified for a better understanding of its structure

2.      The data presented in "2.2.3 Physical-chemical properties" is better to present in a separate table.

Answer: The data was placed on the table as requested.

3.      The brick material should be considered as a porous material not the solid.

Answer: I agree that the brick should be considered as porous material. In the brick should be reported in the article as a solid or porous material domain as pertinent.

4.      "Figure 20. Pressure field" is not true as there is an inlet and outlet with pressure difference.

Answer: The mass flow rate is very slow and it results in very low air pressure variations so that the software cannot capture the pressure fields. To improve visualization, the figure has been replaced by another one that displays the pressure gradients along the X coordinate.

5.      Data in "Figure 7" should be discussed. Why the corners are the hot points and inside are cold?

6.      Answer: The outer corners on the right side heat up more quickly as they are in direct contact with the hot air flowing over the brick. The internal corners of the brick persist with lower temperature because of the difficulty of the air to reach these regions. When it reaches these regions, its velocity is lower.

7.      Detail of thermal measurements of reference 2 is missing.

Answer: The detail of thermal measurements of reference 2 were added to the text.

8.      More references should be added: 

https://doi.org/10.1016/j.jclepro.2014.07.028 https://doi.org/10.1016/j.conbuildmat.2017.09.052 https://doi.org/10.1016/j.conbuildmat.2018.05.285

Answer: As proposed, the three references were inserted in article

All modification occurred in the text are highlighted in red.

Round  2

Reviewer 1 Report

With the changes, the article can be accepted for publication.

Author Response

Dear reviewer,

Thanks for your comments and suggestions. Below there are the answer for all questions.

The text was revised by two experts in the English language. All modification occurred in the text are highlighted in red.

Reviewer 2 Report

When the mass flow rate is very slow the model should be presented by heat conduction and CFD is not useful. This fact should be discussed in research design part.

Author Response

Dear reviewer,

Thanks for your comments and suggestions. The text was revised by two experts in the English language.

1.      When the mass flow rate is very slow the model should be presented by heat conduction and CFD is not useful. This fact should be discussed in research design part.

Answer: The suggested text has been added.

All modification occurred in the text are highlighted in red.
